# Removal of Pb$^{2+}$ from Aqueous Solutions Using K-Type Zeolite Synthesized from Coal Fly Ash

**Yuhei Kobayashi [1], Fumihiko Ogata [1], Chalermpong Saenjum [2,3], Takehiro Nakamura [1] and Naohito Kawasaki [1,4,*]**

[1] Faculty of Pharmacy, Kindai University, 3-4-1 Kowakae, Higashi-Osaka, Osaka 577-8502, Japan; 1944420001t@kindai.ac.jp (Y.K.); ogata@phar.kindai.ac.jp (F.O.); nakamura@phar.kindai.ac.jp (T.N.)

[2] Faculty of Pharmacy, Chiang Mai University, Suthep Road, Muang District, Chiang Mai 50200, Thailand; chalermpong.saenjum@gmail.com

[3] Cluster of Excellence on Biodiversity-Based Economics and Society (B.BES-CMU), Chiang Mai University, Suthep Road, Muang District, Chiang Mai 50200, Thailand

[4] Antiaging Center, Kindai University, 3-4-1 Kowakae, Higashi-Osaka, Osaka 577-8502, Japan

\* Correspondence: kawasaki@phar.kindai.ac.jp; Tel.: +81-6-4307-4012

**Abstract:** In this study, a novel zeolite (K-type zeolite) was synthesized from coal fly ash (FA), and adsorption capacity on Pb$^{2+}$ was assessed. Six types of zeolite (FA1, FA3, FA6, FA12, FA24, and FA48) were prepared, and their physicochemical properties, such as surface functional groups, cation exchange capacity, pH$_{pzc}$, specific surface area, and pore volume, were evaluated. The quantity of Pb$^{2+}$ adsorbed by the prepared zeolites followed the order FA < FA1 < FA3 < FA6 < FA12 < FA24 < FA48. Current results indicate that the level of Pb$^{2+}$ adsorbed was strongly related to the surface characteristics of the adsorbent. Additionally, the correlation coefficient between the amounts of Pb$^{2+}$ adsorbed and K$^+$ released from FA48 was 0.958. Thus, ion exchange with K$^+$ in the interlayer of FA48 is critical for the removal of Pb$^{2+}$ from aqueous media. The new binding energies of Pb(4f) at 135 and 140 eV were detected after adsorption. Moreover, FA48 showed selectivity for Pb$^{2+}$ adsorption in binary solution systems containing cations. The results revealed that FA48 could be useful for removing Pb$^{2+}$ from aqueous media.

**Keywords:** K-type zeolite; fly ash; lead; adsorption

## 1. Introduction

In recent years, issues faced by the aquatic environment, such as water wastage (Goal 6) and the presence of plastic bags in the ocean (Goal 14), have become major concerns for sustainable societal development [1]. Several heavy metals have been contaminating the aquatic environment through human activities, which are non-biodegradable and accumulate in the ecosystem via the food chain, causing various health problems and diseases [2]. Among these heavy metals, lead (Pb$^{2+}$), mercury (Hg$^{2+}$), and cadmium (Cd$^{2+}$) have been defined as the "big three" harmful heavy metals posing the greatest threat to humans, animals, and the environment [3,4]. Numerous studies have previously identified a relationship between Pb$^{2+}$ exposure and human health impacts, such as neurotoxicity, cardiovascular problems, kidney damage, hormonal imbalances, and decreased musculoskeletal function [5,6]. Pb$^{2+}$ has been classified by the International Agency for Research on Cancer as Group 2B which is possibly carcinogenic to humans. Additionally, drinking water problems involving Pb$^{2+}$ have occurred in several countries [7–10]. Therefore, the World Health Organization and U.S. Environmental Protection Agency have established maximum permissible Pb$^{2+}$ contents for drinking water of 0.01 and 0.015 mg L$^{-1}$, respectively. Thus, the removal of Pb$^{2+}$ from aqueous media is an important global issue.

In Japan, the demand for coal-fired power plants increased after the Fukushima Daiichi Nuclear Power Station accident in 2001. Coal fly ash, a byproduct of coal combustion, is produced from coal-fired power plants (approximately 11.5 million tons in Japan in 2017) [11]. Approximately 750 million tons of coal fly ash are produced globally, and only 25% of this ash is recycled in cement, concrete, soil conditioner, and fertilizer materials [12–14]. Therefore, a recycling technology for coal fly ash must be developed to utilize its unused value [14]. $SiO_2$, $Al_2O_3$, and $Fe_2O_3$ were identified as the major chemical components of coal fly ash. Various studies have previously reported the adsorption capability of heavy metals using coal fly ash; however, they suggested a lower adsorption capacity than that of conventional adsorbents [15,16].

Different activation methods have been modified and developed to improve the adsorption capacity of coal fly ash [17]. Such methods include the conversion of coal fly ash to zeolite named hydrothermal activation, which is useful for decreasing the amount of coal fly ash and preparing an adsorbent with a high adsorption capacity for heavy metals. Zeolite is a general term for crystalline hydrous aluminosilicate, which consists of a three-dimensional network structure of $SiO_4$–$AlO_4$ tetrahedrons with Si and Al as basic constituents. Their structures contain channels and cavities of different sizes with unique physicochemical properties, such as their adsorption, ion exchange, molecular sieving, and catalytic abilities [18–21]. Additionally, the physicochemical properties of zeolite prepared from coal fly ash following the activation using hydrothermal technique are directly affected by certain parameters, such as the coal fly ash composition, alkaline solution concentration, reaction temperature, time, and pressure, volume ratio of the alkaline solution, and amount of coal fly ash [14,22–24]. Therefore, suitable conditions for preparing zeolite from coal fly ash generated by coal-fired power plants in Japan must be identified. Our previous studies elucidated the composition of coal fly ash generated from the Tachibana-Wan Thermal Power Station in Japan [25]. Moreover, Na-type zeolite could be produced from coal fly ash, and adsorption capability on heavy metals was studied and evaluated [26]. Additionally, various previous studies have also investigated the adsorption of $Pb^{2+}$ from aqueous media using industrial/agriculture wastes such as wool, sawdust, sugar beet pulp, and *Azadirachta indiea* (Neem) leaf [27–30]. However, the physicochemical properties of other types of zeolite produced from coal fly ash generated by the Tachibana-wan Thermal Power Station and its heavy metals adsorption capacity have not yet been reported.

Sodium hydroxide is often used in the conversion of FA into zeolite by hydrothermal activation, and the ion in the prepared zeolite can then be exchanged with sodium ions in aqueous media. Some studies have already reported the synthesis and characterization of zeolite prepared from FA using a sodium hydroxide solution [12,18,19,23,24,26]. However, few have been conducted on the synthesis and physicochemical properties of zeolite prepared from fly ash using potassium hydroxide solution. The preparation of K-type zeolite from coal ash for use as a fertilizer in the agricultural field has been reported [14]. However, those studies did not sufficiently evaluate the adsorption capacity for heavy metals. Additionally, the cation exchange capacity (ion exchange capacity) in zeolite is one of the most important adsorption mechanisms. This indicates that sodium ion (or potassium ion) is exchanged with cation ($Pb^{2+}$ in this study) using Na-type zeolite (or K-type zeolite). The ionic radii of sodium ion and potassium ion are 1.80 and 2.20 Å, respectively. In addition, the (hydrate) radii of lead ion is over 1.8 Å [17,31]. Therefore, it was expected that K-type zeolite would be useful for removal of $Pb^{2+}$ from aqueous solution compared to Na-type zeolite.

Therefore, the aim of this study was to synthesize K-type zeolite from coal fly ash following the activation using hydrothermal technique and evaluate adsorption capability on $Pb^{2+}$ from aqueous media. The influences of various parameters including initial concentration, temperature, pH, contact time, and selectivity were additionally explored, and the $Pb^{2+}$ adsorption mechanism of K-type zeolite was elucidated.

## 2. Materials and Methods

### 2.1. Materials and Chemicals

Standard solutions of $Pb^{2+}$ ($Pb(NO_3)_2$ in 0.1 mol $L^{-1}$ $HNO_3$), $Na^+$ (NaCl in water), $Mg^{2+}$ ($Mg(NO_3)_2$ in 0.1 mol $L^{-1}$ $HNO_3$), $K^+$ (KCl in water), $Ca^{2+}$ ($CaCO_3$ in 0.1 mol $L^{-1}$ $HNO_3$), $Ni^{2+}$ ($Ni(NO_3)_2$ in 0.1 mol $L^{-1}$ $HNO_3$), $Cu^{2+}$ ($Cu(NO_3)_2$ in 0.1 mol $L^{-1}$ $HNO_3$), $Zn^{2+}$ ($Zn(NO_3)_2$ in 0.1 mol $L^{-1}$ $HNO_3$), $Sr^{2+}$ ($SrCO_3$ in 0.1 mol $L^{-1}$ $HNO_3$), and $Cd^{2+}$ ($Cd(NO_3)_2$ in 0.1 mol $L^{-1}$ $HNO_3$), potassium hydroxide, nitric acid, and sodium hydroxide were purchased from FUJIFILM Wako Pure Chemical Co., Osaka, Japan. Coal fly ash (FA, JIS Type-II) was obtained from the Tachibana-wan Thermal Power Station (Shikoku Electric Power, Inc., Takamatsu, Japan). K-type zeolites were produced following a previously reported method [32]. Briefly, FA (3.0 g) was mixed with 3-mol $L^{-1}$ potassium hydroxide solution in the volume of 240 mL. The reaction solution was then shaken and heated at 93 °C for 1, 3, 6, 12, 24, and 48 h. Afterwards, the suspensions were filtered through a 0.45-μm membrane filter (Advantec MFS, Inc., Tokyo, Japan), and obtained samples were washed with distilled water and then dried at 50 °C for 1 day. The samples heated for different durations were denoted as FA1, FA3, FA6, FA12, FA24, and FA48, respectively.

### 2.2. Physicochemical Properties of the Adsorbents

The physicochemical properties of each adsorbent were analyzed as follows. The crystal structure and morphology were analyzed using a MiniFlex II (Rigaku, Tokyo, Japan) and SU1510 (Hitachi Ltd., Tokyo, Japan), respectively. The specific surface area and pore volume were measured using a NOVA4200*e* instrument (Yuasa Ionic, Kyoto, Japan). The cation exchange capacity (CEC) and $pH_{pzc}$ were determined following the Japanese Industrial Standard Method (JIS K 1478: 2009) and the method reported by Faria et al. [33]. The concentrations of acidic or basic functional groups in the adsorbents were measured following the Boehm titration method [34]. Additionally, the binding energy was analyzed using an AXIS-NOVA instrument (Shimadzu Co., Ltd., Kyoto, Japan). Finally, the solution pH was measured using an F-73S digital pH meter (HORIBA, Ltd., Kyoto, Japan).

### 2.3. Adsorption Capacity of $Pb^{2+}$

Each adsorbent (0.01 g) was mixed with 50 mL of the $Pb^{2+}$ solution at 50 mg $L^{-1}$ (pH 3.0). The reaction solution was shaken at 100 rpm and 25 °C for 24 h, and then filtered through a 0.45-μm membrane filter. The concentration of $Pb^{2+}$ was measured using an iCAP-7600 Duo (Thermo Fisher Scientific Inc., Tokyo, Japan). The quantity of $Pb^{2+}$ adsorbed was calculated as the difference between the $Pb^{2+}$ concentrations before and after adsorption.

### 2.4. Effect of Initial Concentration, Temperature, pH, Contact Time, and Coexisting Ions on $Pb^{2+}$ Adsorption

First, 0.01 g of FA48 was mixed with a 50-mL $Pb^{2+}$ solution at concentrations of 10, 20, 30, 40, and 50 mg $L^{-1}$ (the concentration range was determined by ecological risk assessment of sediment and total concentrations of heavy metals in sewage sludge from wastewater discharging area) [35–37]. The reaction solution was shaken at 100 rpm and 7, 25, and 45 °C for 24 h using water bath shaker MM-10 (TAITEC Co., Nagoya, Japan), and then filtered through a 0.45-μm membrane filter. Second, to evaluate the pH, 0.01 g of FA48 was mixed with 50 mL of a $Pb^{2+}$ solution at concentrations of 10, 30, and 50 mg $L^{-1}$. The pH of the solution was adjusted between 2, 3, 5, 7, and 9 using nitric acid or sodium hydroxide solutions. The reaction solution was then shaken at 100 rpm and 25 °C for 24 h. To evaluate the contact time, 0.01 g of FA48 was mixed with 50 mL of a $Pb^{2+}$ at a concentration of 50 mg $L^{-1}$. The reaction solution was then shaken at 100 rpm and 25 °C for 0.5, 1, 3, 6, 18, 21, 24, 30, 42, and 48 h. The quantity of $Pb^{2+}$ adsorbed in each case was calculated as described above, and the data are presented as the mean ± standard deviation. Additionally, to evaluate the adsorption mechanism, the quantity of $K^+$ released from FA48 in the adsorption isotherm experiment was also measured using

an iCAP-7600 Duo device. Finally, to evaluate the $Pb^{2+}$ adsorption selectivity of the zeolite, 0.01 g of FA48 was mixed with 50 mL of a binary solution system at a concentration of 10 mg $L^{-1}$ ($Pb^{2+}$ and $Na^+$, $Mg^{2+}$, $K^+$, $Ca^{2+}$, $Ni^{2+}$, $Cu^{2+}$, $Zn^{2+}$, $Sr^{2+}$, or $Cd^{2+}$). The reaction solution was shaken at 100 rpm and 25 °C for 24 h. The concentration of each metal was measured using an iCAP-7600 Duo device. The quantity of each metal adsorbed was also calculated based on the differences in concentration before and after adsorption.

## 3. Results and Discussion

### 3.1. Physicochemical Properties

Zeolite is a highly porous aluminosilicate containing various channels and cavities. Ion-exchangeable cations, such as sodium, potassium, and calcium were used to balance the negative charge [21,38]. The scanning electron microscopy (SEM) images of each zeolite are shown in Figure 1. Spherical particles with different diameters were observed in FA. The spheres and agglomerates of FA1, FA3, FA6, and FA12 could be maintained, and some changes on the surfaces of each type of FA were observed under the tested experimental conditions. The spheres and agglomerates of FA24 and FA48 significantly changed to different crystal shapes and their particle sizes decreased. The synthesis of zeolite from fly ash involves three steps, i.e., dissolution, condensation, and crystallization [39]. Similar trends were observed in previous studies [12,17].

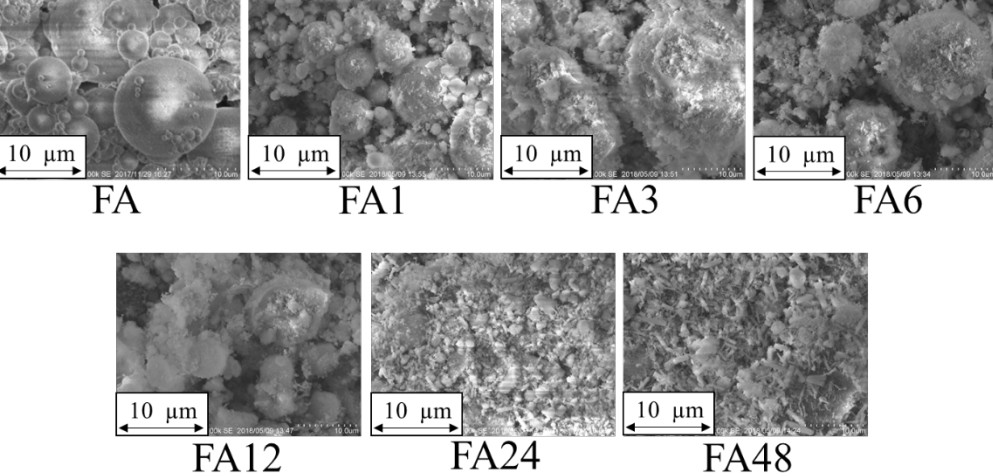

**Figure 1.** SEM images of adsorbents. Magnification is 3000 diameters.

The X-ray diffraction (XRD) patterns of the samples are shown in Figure 2. The FA was composed of quartz ($SiO_2$) and mullite ($3Al_2O_3 \cdot 2SiO_2$), and no significant changes in the XRD patterns of FA, FA1, FA3, FA6, and FA12 under our experimental conditions were observed. However, the structure of Zeolite F (($K_{13.5}Si_{10}Al_{10}O_{40}$)(OH)$_3 \cdot 13H_2O$) appeared in FA24 and FA48, indicating that FA was converted into zeolite. Less time was required to convert FA into zeolite using sodium hydroxide (approximately 6 h) than that required when using potassium hydroxide (approximately 24 h). Trends similarly were observed in a previous study [14]. The physicochemical properties of the adsorbents are shown in Table 1, and the concentrations of basic functional groups of FA24 and FA48 were higher than those of the other adsorbents. However, the concentrations of acidic functional groups decreased with raising the time of alkaline activation. The $pH_{pzc}$ were not significantly different between the different types of FAs in this study. The CEC and pore volume ($d \leq 20$ (Å)), which greatly influence the adsorption capacity, of FA24 and FA48 were 12.4–26.4 times and 100 times higher than those of other FAs, indicating that the hydrothermal activation method significantly affected the CEC and pore volume ($d \leq 20$ (Å)) under our experimental conditions. Additionally, the specific surface area and pore volume (such as $20 < d \leq 500$ (Å), total, and mean pore diameter) of FA6 and FA12 were higher

than those of the other FAs. Our previous study reported the characteristics of Na-type zeolite [26], and the CEC and pore volume ($d \leq 20$ (Å)) of the K-type zeolite prepared in this study exceeded those of the Na-type zeolite. Thus, hydrothermal activation using FA and potassium hydroxide was useful for producing novel zeolites to remove metals from aqueous media.

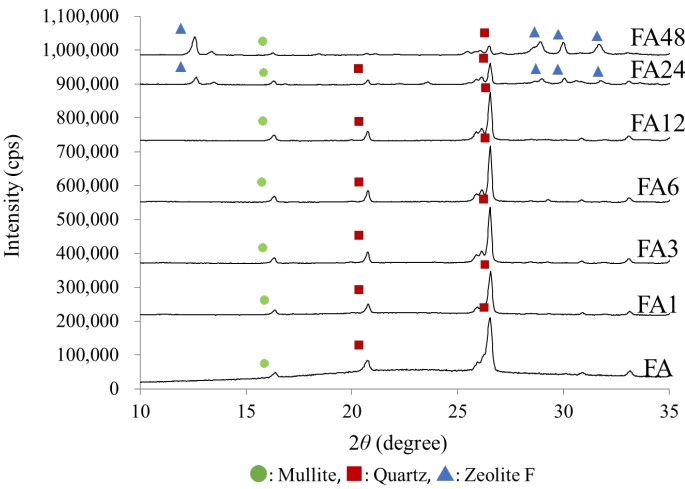

**Figure 2.** XRD patterns of adsorbents.

**Table 1.** Physicochemical properties of adsorbents.

| Adsorbents | | FA | FA1 | FA3 | FA6 | FA12 | FA24 | FA48 |
|---|---|---|---|---|---|---|---|---|
| Concentration of basic functional groups (mmol g$^{-1}$) | | 0 | 0.41 | 0.77 | 1.38 | 1.36 | 1.48 | 1.43 |
| Concentration of acidic functional groups (mmol g$^{-1}$) | | 0.10 | 0.24 | 0.30 | 0.31 | 0.31 | 0.24 | 0.24 |
| CEC (mmol g$^{-1}$) | pH 5 | 0.34 | 1.98 | 1.17 | 1.63 | 2.27 | 7.90 | 8.98 |
| | pH 10 | 0.19 | 0.65 | 1.55 | 2.09 | 3.46 | 11.17 | 11.17 |
| pH$_{pzc}$ | | 9.8 | 9.3 | 9.3 | 9.5 | 9.7 | 10.4 | 10.4 |
| Specific surface area (m$^2$ g$^{-1}$) | | 1.4 | 15.1 | 31.5 | 53.3 | 54.5 | 50.3 | 47.3 |
| Pore volume (μL g$^{-1}$) | $d \leq 20$ (Å) | 0.1 | 0.9 | 0 | 0.5 | 0.2 | 10.0 | 10.0 |
| | $20 < d \leq 500$ (Å) | 2.0 | 41.9 | 97.4 | 161.5 | 185.0 | 105.0 | 99.0 |
| | Total | 2.2 | 63.0 | 139.0 | 221.0 | 220.0 | 151.0 | 131.0 |
| Mean pore diameter (Å) | | 57.0 | 167.2 | 176.7 | 165.9 | 161.6 | 120.1 | 110.7 |

### 3.2. Adsorption Capacity of Pb$^{2+}$

FA, FA1, FA3, FA6, FA12, FA24, and FA48 adsorbed 1.04, 1.45, 4.87, 8.11, 13.54, 54.29, and 55.53 mg g$^{-1}$ of Pb$^{2+}$, respectively. The relationships between the quantity of Pb$^{2+}$ adsorbed and the parameters in Table 1 were statistically assessed (Table 2), and the correlation coefficients value between the level of Pb$^{2+}$ adsorbed and CEC, pH$_{pzc}$, and pore volume ($d \leq 20$ (Å)) were positive, at 0.986–0.999, 0.921, and 0.980, respectively. Therefore, these properties have the greatest influence on the Pb$^{2+}$ adsorption capacity in solution. Additionally, FA24 and FA48 adsorbed more Pb$^{2+}$ than Na-type zeolite (approximately 30 mg g$^{-1}$) under the same experimental conditions [26]. This is because K-type zeolite has a high CEC and pore volume ($d \leq 20$ (Å)) than Na-type zeolite. In this study, FA48 was selected to study and evaluate the adsorption capability on Pb$^{2+}$ in the following experiments.

### 3.3. Pb$^{2+}$ Adsorption Isotherms

The Pb$^{2+}$ adsorption isotherms are shown in Figure 3. The quantity of Pb$^{2+}$ adsorbed increased with raising initial concentration. Amount adsorbed was 2.0 times (7 °C), 2.36 times (25 °C), and 2.44 times (45 °C) raised from 10 to 50 mg L$^{-1}$ of initial concentration. Additionally, the quantity of Pb$^{2+}$ adsorbed also increased with raising temperature, thereby indicating that Pb$^{2+}$ adsorption using FA48 was

endothermic. In this study, the distribution of $Pb^{2+}$ between the aqueous media and FA48 was described using adsorption isotherms based on a set of assumptions related to the heterogeneity or homogeneity of the FA48 [24]. Thus, two useful isotherm models, i.e., Freundlich and Langmuir, were selected [40,41]. The Freundlich isotherm model is applicable to adsorption onto heterogeneous surfaces, whereas the Langmuir isotherm is applicable to monolayer and homogeneous adsorption [17]. The Freundlich and Langmuir models are described by Equations (1) and (2), respectively:

$$\log q = \frac{1}{n} \log C_e + \log k, \tag{1}$$

$$1/q = 1/(W_s a C_e) + 1/W_s, \tag{2}$$

where $q$ is the quantity of $Pb^{2+}$ adsorbed (mg g$^{-1}$), $C_e$ is the equilibrium concentration (mg L$^{-1}$), and $k$ and $n$ are the adsorption capacity and intensity, respectively. Moreover, $W_s$ is the maximum quantity of adsorbed $Pb^{2+}$ (mg g$^{-1}$), and $a$ is the coefficient reflecting the relative sorption and desorption rates at equilibrium (L mg$^{-1}$). The Freundlich and Langmuir constants for the adsorption of $Pb^{2+}$ are summarized in Table 3. In this study, the isotherm data were fitted to both the Freundlich (correlation coefficient: ≥0.991) and Langmuir (correlation coefficient: ≥0.960) models. The maximum adsorption capacity ($W_s$) of $Pb^{2+}$ increased with raising temperature, thereby indicating that the adsorption of $Pb^{2+}$ by FA48 is an endothermic process (Figure 3). Additionally, when the value of $1/n$ is 0.1–0.5, adsorption readily occurs. However, when the value of $1/n$ exceeds 2, adsorption is difficult [42]. Herein, the value of $1/n$ was 0.30–0.32, and the adsorption of $Pb^{2+}$ using FA48 in aqueous solutions were more favorable.

**Table 2.** Correlation coefficients between quantity adsorbed and physicochemical properties.

| Adsorbents | | FAs |
|---|---|---|
| Concentration of basic functional groups (mmol g$^{-1}$) | | 0.677 |
| Concentration of acidic functional groups (mmol g$^{-1}$) | | 0.044 |
| CEC | pH 5 | 0.986 |
| (mmol g$^{-1}$) | pH 10 | 0.999 |
| pH$_{pzc}$ | | 0.921 |
| Specific surface area (m$^2$ g$^{-1}$) | | 0.549 |
| Pore volume (μL g$^{-1}$) | $d \leq 20$ (Å) | 0.980 |
| | $20 < d \leq 500$ (Å) | 0.201 |
| | Total | 0.227 |
| Mean pore diameter (Å) | | 0.262 |

To evaluate the adsorption mechanism of $Pb^{2+}$ using FA48, the relationships between the quantities of $Pb^{2+}$ and $K^+$ adsorbed by and released from FA48 were explored, respectively (Figure 4A). The correlation coefficient was positive at 0.958 under our experimental conditions, suggesting that ion exchange was involved in the adsorption of $Pb^{2+}$. As mentioned above, the correlation coefficient of the relationship between the quantity of $Pb^{2+}$ adsorbed and the CEC value was 0.986–0.999, demonstrating that ion exchange was involved in the removal capacity on $Pb^{2+}$ from aqueous media. A similar result was achieved using Na-type zeolite in a previous study [26]. Additionally, the binding energies of lead before and after adsorption were measured (Figure 4B). New peaks of Pb(4f) were detected at 135 and 140 eV after adsorption, indicating that $Pb^{2+}$ was adsorbed onto the surface of FA48 in this study. Thus, the surface properties of FA48 are vital in the removal ability on $Pb^{2+}$ from aqueous media, and these results agree with those in Table 2.

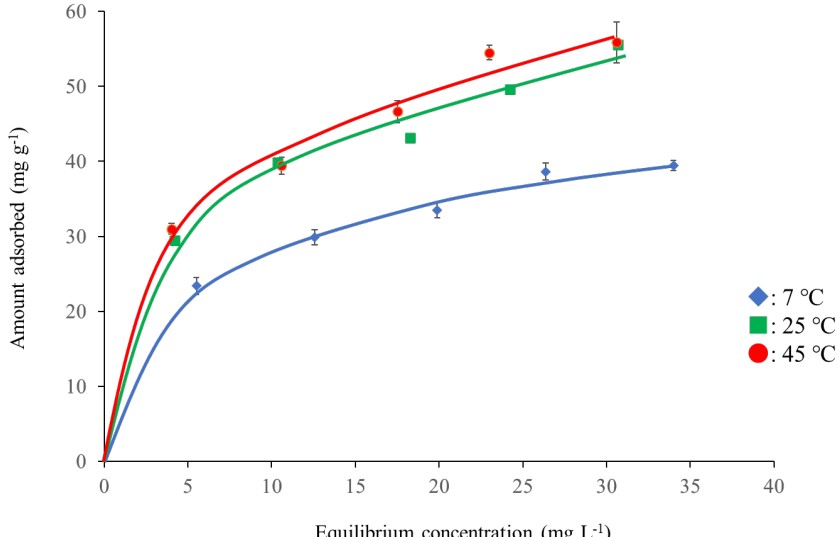

**Figure 3.** Adsorption isotherms of Pb$^{2+}$ using FA48 at different temperatures. Initial concentration 10, 20, 30, 40, and 50 mg L$^{-1}$, solvent volume 50 mL, adsorbent 0.01 g, contact time 24 h.

**Table 3.** Langmuir and Freundlich constants for the adsorption of Pb$^{2+}$.

| Adsorbent | Temperatures (°C) | Langmuir Constants | | | Freundlich Constants | | |
|---|---|---|---|---|---|---|---|
| | | $W_s$ (mg g$^{-1}$) | $a$ (L mg$^{-1}$) | $r$ | log$k$ | $1/n$ | $r$ |
| FA48 | 7 | 74.1 | 0.20 | 0.982 | 1.38 | 0.30 | 0.994 |
| | 25 | 98.0 | 0.24 | 0.987 | 1.51 | 0.30 | 0.997 |
| | 45 | 102.0 | 0.25 | 0.960 | 1.52 | 0.25 | 0.960 |

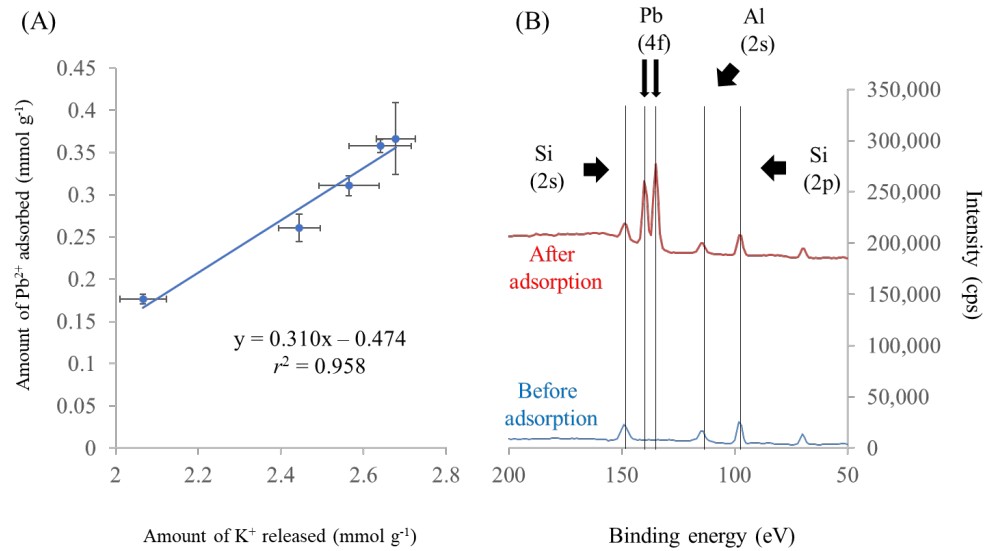

**Figure 4.** Correlation coefficient between quantity of Pb$^{2+}$ adsorbed and quantity of K$^+$ released from FA48 (**A**) and binding energy of lead onto FA48 surface before and after adsorption (**B**).

### 3.4. Effect of pH on Pb$^{2+}$ Adsorption

The solution pH is critical for determining the capacity of Pb$^{2+}$ adsorption from aqueous media. Figure 5 exerts the effect of pH on the adsorption of Pb$^{2+}$. The quantity adsorbed increased as the pH raised from 2 to 5 under our experimental conditions. Above pH 5, the adsorption capacity decreased

sharply. Under acidic conditions, the FA48's surface will be covered with protons ($H^+$), with which $Pb^{2+}$ competes for adsorption sites. A previous study reported that zeolite could preferentially adsorb $H^+$ from aqueous media also containing heavy metal ions [43]. The edge groups with a positive charge ($Al\text{-}OH_2^+$ and $Si\text{-}OH_2^+$) on the FA48 surface are explained below [44].

$$-AlOH + H_2O\ (H^+ + OH^-) \rightarrow -AlOH_2^+ + OH^-$$
$$-SiOH + H_2O\ (H^+ + OH^-) \rightarrow -SiOH_2^+ + OH^- \tag{3}$$

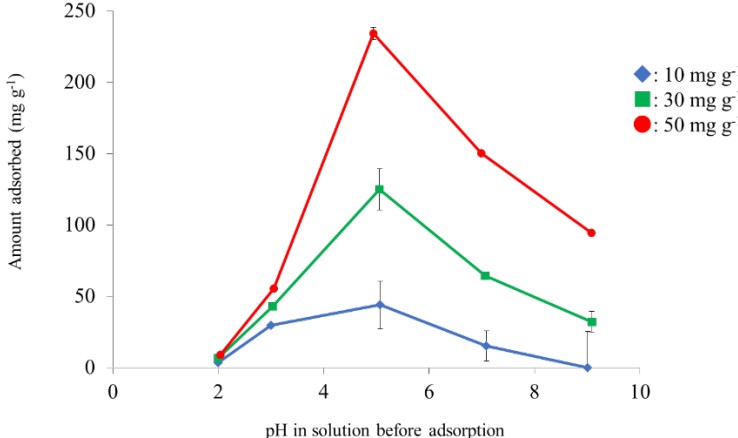

**Figure 5.** Effect of pH on the adsorption of $Pb^{2+}$ onto FA48. Initial concentration 10, 30, and 50 mg $L^{-1}$, solvent volume 50 mL, adsorbent 0.01 g, pH 2, 3, 5, 7, and 9, contact time 24 h, temperature 25 °C, agitation speed 100 rpm.

Thus, electrostatic repulsion easily occurred between them, and it was also difficult for ion exchange to occur between $Pb^{2+}$ and $K^+$ in aqueous media. As the pH increased, the deprotonation of hydroxyl groups increased, indicating that the number of anionic sites (adsorption sites) on FA48 increased, thereby contributing to the adsorption of $Pb^{2+}$ [17]. However, this did not increase the removal percentage of $Pb^{2+}$ beyond pH 6.0 in our experimental condition. The forms of $Pb^{2+}$ in the solution are $Pb^{2+}$, $Pb(OH)^+$, $Pb(OH)_2$, $Pb(OH)_3^-$, $Pb(OH)_4^{2-}$, $Pb_2(OH)_3^+$, and $Pb_3(OH)_4^{2+}$ [45], and the predominant species of $Pb^{2+}$ at pH < 7 are $Pb^{2+}$ and $Pb(OH)^+$, whereas $Pb(OH)_2$, $Pb(OH)_3^-$, and $Pb(OH)_4^{2-}$ are the main forms of $Pb^{2+}$ at pH > 7–8. Thereby the complexion between free $Pb^{2+}$ available and FA48 surface was not occurred in our experimental condition. Similar trend was reported by a previous study [46]. Additionally, there is deprotonation as basicity increases beyond pH 6.0 [46], suggesting that electron repulsion occurred between the dissolved negatively charged Pb species and the negatively charged surface of FA48. Thus, the $Pb^{2+}$ adsorption capacity decreased under basic conditions. However, further studies are necessary for elucidate the effect of pH on the adsorption of $Pb^{2+}$ using FA48 from aqueous media in detail.

### 3.5. Effect of Contact Time on $Pb^{2+}$ Adsorption

The effect of contact time on the adsorption capability of $Pb^{2+}$ is shown in Figure 6. The adsorption equilibrium was attained within approximately 6 h under our experimental conditions. Kinetic experiments were conducted to indicate the rate of adsorption and the potential of the rate-limiting step [2]. The pseudo-first-order (Equation (4)) and pseudo-second-order (Equation (5)) models were employed to evaluate these factors and can be described as follows [47,48].

$$\ln(q_{e,exp} - q_t) = \ln q_{e,cal} - k_1 t, \tag{4}$$

$$\frac{t}{q_t} = \frac{t}{q_{e,cal}} + \frac{1}{k_2 \times q_{e,cal}^2}, \tag{5}$$

where $q_{e,exp}$ and $q_{e,cal}$ are the quantities of $Pb^{2+}$ adsorbed in the experiment and calculation (mg g$^{-1}$), and $k_1$ and $k_2$ are the pseudo-first-order (h$^{-1}$) and pseudo-second-order (g mg$^{-1}$ h$^{-1}$) rate constants, respectively. The rapid adsorption during the initial stage was due to the availability of adsorption sites on FA48. Afterwards, the adsorption equilibrium may have been reached due to the limited mass transfer of $Pb^{2+}$ from the bulk liquid phase to the external surface of FA48 [24]. Similar trends were reported in previous studies [49,50]. Table 4 summarizes the kinetic constants of the pseudo-first-order and second-order models. The kinetic data fitted to the pseudo-second-order model (correlation coefficient: 0.998) better than the pseudo-first-order model (correlation coefficient: 0.602). Therefore, the adsorption mechanism of $Pb^{2+}$ using FA48 in this study may be chemisorption. Additionally, the value of $q_{e,exp}$ (99.5 mg g$^{-1}$) was similar to that of $q_{e,cal}$ (98.0 mg g$^{-1}$) in the pseudo-second-order model. Thus, chemical adsorption, one of the adsorption mechanisms, could be the rate-limiting step in the adsorption of $Pb^{2+}$ using FA48 [3].

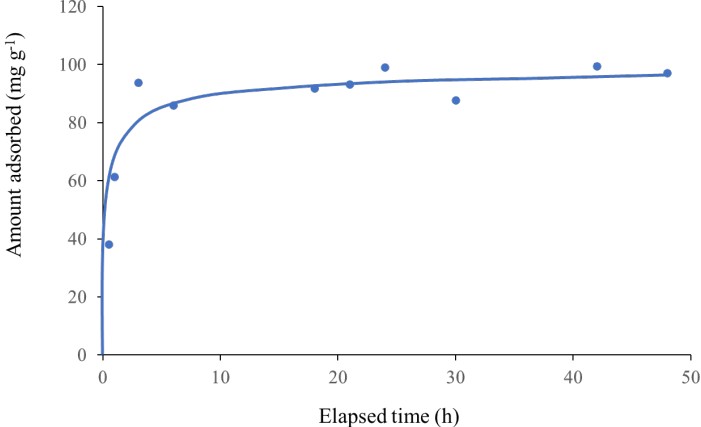

**Figure 6.** Effect of contact time on the adsorption of $Pb^{2+}$ using FA48. Initial concentration 50 mg L$^{-1}$, solvent volume 50 mL, adsorbent 0.01 g, contact time 0.5, 1, 3, 6, 18, 21, 24, 30, 42, and 48 h, temperature 25 °C, agitation speed 100 rpm.

**Table 4.** Kinetic parameters for the adsorption of $Pb^{2+}$.

| Adsorbent | $q_{e,exp}$ | Pseudo-First-Order Model | | | Pseudo-Second-Order Model | | |
|---|---|---|---|---|---|---|---|
| | | $k_1$ (h$^{-1}$) | $q_{e,cal}$ (mg g$^{-1}$) | $r$ | $k_2$ (g mg$^{-1}$ h$^{-1}$) | $q_{e,cal}$ (mg g$^{-1}$) | $r$ |
| FA48 | 99.5 | 0.057 | 19.7 | 0.602 | 0.010 | 98.0 | 0.998 |

### 3.6. Effect of Coexisting Ions on $Pb^{2+}$ Adsorption

In this study, the effect of coexisting ions on the adsorption capability of $Pb^{2+}$ using FA48 was demonstrated in Table 5. $Na^+$, $Mg^{2+}$, $K^+$, $Ca^{2+}$, $Ni^+$, $Cu^{2+}$, $Zn^{2+}$, $Sr^{2+}$, and $Cd^{2+}$ were selected as the components of the binary solution system, which exist in aquatic environments or wastewater from factories (human activities) [38,51]. The adsorption percentage of $Pb^{2+}$ using FA48 in a single solution system was 75.7%. The $Pb^{2+}$ adsorption selectivity of FA48 in a binary solution system (adsorption percentage was approximately >60.7%) exceeded that of other cations under our experimental conditions. These results indicate that the adsorption capacity of FA48 was influenced by cationic factors, such as charge density and hydrate ion diameter, and the accessibility of the active sites of FA48 [51]. The radius of hydrated ions and electronegativity of $Pb^{2+}$ exceed those of other cations, such as $Ni^{2+}$, $Cu^{2+}$, $Cd^{2+}$, and $Zn^{2+}$. Similar trends were reported in previous studies [38,51].

Additionally, the adsorption capacity of cations excluding $Pb^{2+}$ was significantly lower than that of $Pb^{2+}$ in this study. Thus, FA48 is useful for the selective removal of $Pb^{2+}$ from aqueous media.

**Table 5.** Adsorption capacity of $Pb^{2+}$ in binary solution system.

| Components in Binary Solution | Removal Percentage of $Pb^{2+}$ (%) | Removal of Other Cations (%) |
|:---:|:---:|:---:|
| $Pb^{2+} + Na^+$ | 60.7 | 0.2 |
| $Pb^{2+} + Mg^{2+}$ | 69.4 | 0 |
| $Pb^{2+} + K^+$ | 75.7 | 0 |
| $Pb^{2+} + Ca^{2+}$ | 62.4 | 0 |
| $Pb^{2+} + Ni^+$ | 73.2 | 0 |
| $Pb^{2+} + Cu^{2+}$ | 62.7 | 3.7 |
| $Pb^{2+} + Zn^{2+}$ | 63.7 | 0.9 |
| $Pb^{2+} + Sr^{2+}$ | 70.0 | 0.7 |
| $Pb^{2+} + Cd^{2+}$ | 64.4 | 4.1 |

## 4. Conclusions

In this study, K type-zeolite (FA48) was synthesized from coal fly ash. The CEC, $pH_{pzc}$, and pore volume ($d \leq 20$ (Å)) of FA48 were higher than those of other zeolites. Additionally, the quantity of $Pb^{2+}$ adsorbed using FA48 (55.53 mg g$^{-1}$) exceeded that of other zeolites. The adsorption of $Pb^{2+}$ was related to physicochemical properties, such as the CEC (0.986–0.999), $pH_{pzc}$ (0.921), and pore volume ($d \leq 20$ (Å)) (0.980). The relationship between the amounts of $Pb^{2+}$ and $K^+$ adsorbed by and released from FA48 was 0.958, respectively. Thus, ion exchange with $K^+$ in the interlayer of FA48 is strongly related to the adsorption capacity. Furthermore, the binding energy of lead at 135 and 140 eV could be detected after adsorption, indicating that the adsorbent's surface characteristics were critical for the removal of $Pb^{2+}$ from aqueous media. The adsorption isotherms and kinetics data demonstrated that the adsorption of $Pb^{2+}$ using FA48 was an endothermic process. Finally, FA48 exhibited selectivity for $Pb^{2+}$ adsorption from a binary solution system containing cations. These results provide useful information for the recycling of coal fly ash and the removal of $Pb^{2+}$ from aqueous media.

**Author Contributions:** Conceptualization, N.K. and F.O.; investigation, Y.K., C.S. and T.N.; writing—original draft preparation, Y.K. and F.O.; writing—review and editing, Y.K., F.O. and C.S.; project administration, N.K. All authors have read and agreed to the published version of the manuscript.

**Funding:** This research received no external funding.

**Conflicts of Interest:** The authors declare no conflict of interest.

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
