# Peer review of "Removal of Pb2+ from Aqueous Solutions Using K-Type Zeolite Synthesized from Coal Fly Ash"

_water, doi:10.3390/w12092375_

Round 1
Reviewer 1 Report
This manuscript brings results of synthesis of new zeolites and their applicability as adsorbents for Pb removal from aqueous systems. Although the manuscript presents an important area of research and the findings are promising, there are several points which, in my opinion, should be addressed before the manuscript is accepted for publication:
General comments
1) Introduction section must be improved by critical evaluation of additional literature on removal of Pb by adsorption using other types of adsorbents produced from industrial/agricultural wastes. Authors have previously published (reference 26) a similar study where a Na-type zeolite was synthesized from the same coal fly ash and used for removal of Pb. Therefore, authors must indicate in the present manuscript the reason of replacing the NaOH activation with KOH. What where the presumed benefits expected from the K-type zeolite, in comparison with the Na-type zeolite?
2) Please explain in the manuscript the rationale of the selection of 10-50 mg/L Pb concentration. Is this concentration range relevant for wastewaters? Or for contaminated natural waters? Please add some references to support your explanation.
Specific comments
Lines 46-47:”after the Fukushima Daiichi Nuclear Power Station in 2001”
Please add the word ”accident”: ”after the Fukushima Daiichi Nuclear Power Station accident in 2001”
Lines 74-75:” The influences of temperature, pH, contact time, and selectivity were additionally explored”
As I see, the influence of initial concentration was also studied. Please mention also this parameter!
Line 101: Section ”2.3. 2.3. Adsorption Capacity of Pb2+”
Please give here also the pH of Pb2+solution used to determine the adsorption capacity!
Line 107: Section ”2.4. Effect of Temperature, pH, and Contact Time on Pb2+ Adsorption”
As I see, in this section initial concentration of Pb was also varied between 10-50 mg/L. You should mention this in the title of the section. And you must discuss the effect of initial concentration in section 3 as a separate subsection!
Line 109: ” The reaction solution was shaken at 100 rpm and 7, 25, and 45 °C for 24”
Please give also the name/manufacturer of the shaker. I am also interested in a shaker which makes possible conducting experiments not only at higher temperatures, but also at lower temperatures than room temperature.
Line 119: Section ”2.5. Effect of Coexisting Ions on Pb2+ Adsorption”
Sections 2.4. and 2.5 must be merged in a single section.
Lines 130-138: ”Sodium hydroxide is often used in the conversion of FA into zeolite…………… Thus, in this study, potassium hydroxide was selected to prepare zeolite following the hydrothermal activation method”
This paragraph must be moved to the ”Introduction” section. See also general comment 1.
Lines 156-158: ”…the acid consumption rates of FA24 and FA48 were higher than those of the other adsorbents. However, the base consumption rate and pHpzc were not significantly different between the different types of FAs in this study”
What is the acid/base consumption rate? They were not defined in your study. I suppose they are related with the number of acidic/basic functional groups, as follows: the acid consumption rates give an estimation of the number of basic functional groups (i.e., the acid consumption rates = the acid consumption required for neutralization of basic functional groups); similarly, the basic consumption rates give an estimation of the number of acidic functional groups. If I am correct, I suggest using the terminology ”number of acidic/basic functional groups” throughout the manuscript, because it was mentioned at section 2.2 among the physicochemical properties determined for the characterization of the adsorbents. Furthermore, I also suggest replacing (throughout the manuscript) the ”number” of acidic/basic functional groups with ”concentration” of acidic/basic functional groups”.
Also, If my assumption (the basic consumption rates = the number of acidic functional groups) is correct then, theoretically, in Table 1, the basic consumption rates should constantly decrease with increasing the time of alkaline activation. Please double check!
Lines 181-183: ”Based on these results, FA48 was selected to evaluate Pb2+ adsorption in the following experiments”
This selection is not the right one. The Pb adsorption capacities of FA24 and FA48 are 54.29 and 55.53, respectively. The difference between them is insignificant. Therefore, from an economic standpoint (chemical activation with KOH, at 93 C, is not cheap), FA24 should have been selected for further experiments.
Lines 245-247: ”These results suggest that electron repulsion occurred between Pb2+ and the negatively charged surface of FA48. Thus, the Pb2+ adsorption capacity decreased under basic conditions”
This is not true! First, because electron ATTRACTION occurs between Pb2+ and a negatively charged surface. Second, because the pHpzc value of 10.4 indicates that at pH ˂ 10.4 the global surface charge of FA48 is ALWAYS positive! However, with increasing pH over the 2-9 range the number of negative centers is increasing. Because, over the pH range of 2-9, the main Pb species are cations of Pb2+ and Pb(OH)+ (reference 37), this means that adsorption of Pb should also increase with increasing pH from 2 to 9. In addition, at alkaline pH of 9 precipitation of Pb is also occuring. All these should, theoretically, lead to one single result: the continuous increase of efficiency of Pb removal with increasing pH over the 2-9 range. You must explain at this section the reason why this is not happening at pH > 5!
Lines 247-248: ”Additionally, the pHpzc value of FA48 was 10.4, which further supports the above-mentioned results”
As mentioned at my previous comment, the pHpzc 10.4 value of FA48 does not TOTALLY support the above-mentioned results! This section must be rewritten!
Line 268: ”were reported in a previous study”
Please correct: ”were reported in previous studies”
Lines 271-274: ”Therefore, the adsorption mechanism of Pb2+ using FA48 in this study may be chemisorption……….Thus, chemical adsorption, such as ion exchange, could be the rate-limiting step in the adsorption of Pb2+ using FA48”.
You must decide which the mechanism of Pb removal is! Chemisorption or ion exchange? Because ion exchange is not a type of chemical adsorption! they are two different sorption processes!
Reviewer 2 Report
The authors describe the synthesis of K-type Zeolite from Coal Fly Ash and it use as a adsorbent for Pb removal from water. The work is peformed well although it is not vey innovative as the fly ash has already been studied for this purpose. However, some interesting results related to K-type zeolites are obtained. The manuscript can benefit if minor changes described below are implemented:
- Page 6, line 204: typo »0.1.0.5«
- Figure 5: more experimental points are needed between pH 2 and pH5
- The section 3.4. is unclear:
- The equations are not correctly written (elements are not balanced)
- There might be something wrong in this sentence »As the pH increased, the dissociation of hydroxyl groups increased, indicating that the number of anionic sites (adsorption sites) on FA48 242 increased« How can hydroxyl groups dissoviate at high pH. Maybe the authors think that they deprotonate.
- Although I know what the authors want to say i suggest to rewrite the sentence »These results suggest that electron repulsion occurred between Pb2+ and the negatively charged surface of FA48« As it is now, the sentence says that there is repulsion between a positive and negative charge. Explain that the dissolved negatively charged Pb- species repulse from the negatively charged surface.
After this minor issues are corrected I support publication of the manuscript.
Round 2
Reviewer 1 Report
No comments